# A Metadata-Driven Approach to Understand Graph Neural Networks

**Ting Wei Li**
University of Michigan
`tingwl@umich.edu`

**Qiaozhu Mei**
University of Michigan
`qmei@umich.edu`

**Jiaqi Ma**
University of Illinois Urbana-Champaign
`jiaqima@illinois.edu`

## Abstract

Graph Neural Networks (GNNs) have achieved remarkable success in various applications, but their performance can be sensitive to specific data properties of the graph datasets they operate on. Current literature on understanding the limitations of GNNs has primarily employed a *model-driven* approach that leverages heuristics and domain knowledge from network science or graph theory to model the GNN behaviors, which is time-consuming and highly subjective. In this work, we propose a *metadata-driven* approach to analyze the sensitivity of GNNs to graph data properties, motivated by the increasing availability of graph learning benchmarks. We perform a multivariate sparse regression analysis on the metadata derived from benchmarking GNN performance across diverse datasets, yielding a set of salient data properties. To validate the effectiveness of our data-driven approach, we focus on one identified data property, the degree distribution, and investigate how this property influences GNN performance through theoretical analysis and controlled experiments. Our theoretical findings reveal that datasets with a more balanced degree distribution exhibit better linear separability of node representations, thus leading to better GNN performance. We also conduct controlled experiments using synthetic datasets with varying degree distributions, and the results align well with our theoretical findings. Collectively, both the theoretical analysis and controlled experiments verify that the proposed metadata-driven approach is effective in identifying critical data properties for GNNs.

## 1   Introduction

Graph Neural Networks (GNNs), as a broad family of graph machine learning models, have gained increasing research interests in recent years. However, unlike the ResNet model [14] in computer vision or the Transformer model [36] in natural language processing, there has not been a dominant GNN architecture that is universally effective across a wide range of graph machine learning tasks. This may be attributed to the inherently diverse nature of graph-structured data, which results in the GNN performance being highly sensitive to specific properties of the graph datasets. Consequently, GNNs that demonstrate high performance on certain benchmark datasets often underperform on others with distinct properties. For example, early GNNs have been shown to exhibit degraded performance when applied to non-homophilous graph datasets, where nodes from different classes are highly interconnected and mixed [45, 46, 32, 11, 9].

However, it is non-trivial to identify and understand critical graph data properties that are highly influential on GNN performance. Current literature primarily employs what we term as a *model-driven* approach, which attempts to model GNN performance using specific heuristics or domain knowledge derived from network science or graph theory [41, 45]. Although this approach can offer an in-depth understanding of GNN performance, it can also be time-consuming, subjective, and may not fully capture the entire spectrum of relevant data properties.

37th Conference on Neural Information Processing Systems (NeurIPS 2023).

To address these limitations and complement the model-driven approach, we propose a *metadata-driven approach* to identify critical data properties affecting GNN performance. With the increasing availability of diverse benchmark datasets for graph machine learning [16, 27], we hypothesize that critical graph data properties can be inferred from the benchmarking performance of GNNs on these datasets, which can be viewed as the metadata of the datasets. More concretely, we carry out a multivariate sparse regression analysis on the metadata obtained from large-scale benchmark experiments [27] involving multiple GNN models and a variety of graph datasets. Through this regression analysis, we examine the correlation between GNN performance and the data properties of each dataset, thereby identifying a set of salient data properties that significantly influence GNN performance.

To validate the effectiveness of the proposed metadata-driven approach, we further focus on a specific salient data property, degree distribution, identified from the regression analysis, and investigate the mechanism by which this data property affects GNN performance. In particular, our regression analysis reveals a decline in GNN performance as the degree distribution becomes more imbalanced. We delve deeper into this phenomenon through a theoretical analysis and a controlled experiment.

We initiate our investigation with a theoretical analysis of the GNN performance under the assumption that the graph data is generated by a Degree-Corrected Contextual Stochastic Block Model (DC-CSBM). Here, we define DC-CSBM by combining and generalizing the Contextual Stochastic Block Model [4] and the Degree-Corrected Stochastic Block Model [17]. Building upon the analysis by Baranwal et al. [3], we establish a novel theoretical result on how the degree distribution impacts the linear separability of the GNN representations and subsequently, the GNN performance. Within the DC-CSBM context, our theory suggests that more imbalanced degree distribution leads to few nodes being linearly separable in their GNN representations, thus negatively impacting GNN performance.

Complementing our theoretical analysis, we conduct a controlled experiment, evaluating GNN performance on synthetic graph datasets with varying degree distribution while holding other properties fixed. Remarkably, we observe a consistent decline in GNN performance correlating with the increase of the Gini coefficient of degree distribution, which reflects the imbalance of degree distribution. This observation further corroborates the findings of our metadata-driven regression analysis.

In summary, our contribution in this paper is two-fold. Firstly, we introduce a novel metadata-driven approach to identify critical graph data properties affecting GNN performance and demonstrate its effectiveness through a case study on a specific salient data property identified by our approach. Secondly, we develop an in-depth understanding of how the degree distribution of graph data influences GNN performance through both a novel theoretical analysis and a carefully controlled experiment, which is of interest to the graph machine learning community in its own right.

## 2   Related Work

### 2.1   Analysis on the Limitations of GNNs

There has been a wealth of existing literature investigating the limitations of GNNs. However, most of the previous works employ the model-driven approach. Below we summarize a few well-known limitations of GNNs while acknowledging that an exhaustive review of the literature is impractical. Among the limitations identified, GNNs have been shown to be sensitive to the extent of homophily in graph data, and applying GNNs to non-homophilous data often has degraded performance [1, 9, 23, 46, 45]. In addition, over-smoothing, a phenomenon where GNNs lose their discriminative power with deeper layers [20, 34, 6], is a primary concern particularly for node-level prediction tasks where distinguishing the nodes within the graph is critical. Further, when applied to graph-level prediction tasks, GNNs are limited by their ability to represent and model specific functions or patterns on graph-structured data, an issue often referred to as the expressiveness problem of GNNs. [41, 30, 25, 43]. Most of these limitations are understood through a *model-driven* approach, which offers in-depth insights but is time-consuming and highly subjective. In contrast, this paper presents a *metadata-driven* approach, leveraging metadata from benchmark datasets to efficiently screen through a vast array of data properties.

## 2.2 Data-Driven Analysis in Graph Machine Learning

With the increasing availability of graph learning benchmarks, there have been several recent studies that leverage diverse benchmarks for data-driven analysis. For example, Liu et al. [24] presents a principled pipeline to taxonomize benchmark datasets. Specifically, by applying a number of different perturbation methods on each dataset and obtaining the sensitivity profile of the resulting GNN performance on perturbed datasets, they perform hierarchical clustering on these sensitivity profiles to cluster statistically similar datasets. However, this study only aims to categorize datasets instead of identifying salient data properties that influence GNN performance. Ma et al. [27] establish a Graph Learning Indexer (GLI) library that curates a large collection of graph learning benchmarks and GNN models and conducts a large-scale benchmark study. We obtain our metadata from their benchmarks. Palowitch et al. [31] introduce a GraphWorld library that can generate diverse synthetic graph datasets with various properties. These synthetic datasets can be used to test GNN models through controlled experiments. In this paper, we have used this library to verify the effectiveness of the identified critical data properties.

## 2.3 Impact of Node Degrees on GNN Performance

There have also been a few studies investigating the impact of node degrees on GNNs. In particular, it has been observed that within a single graph dataset, there tends to be an accuracy discrepancy among nodes with varying degrees [35, 22, 44, 39]. Typically, GNN predictions on nodes with lower degrees tend to have lower accuracy. However, the finding of the Gini coefficient of the degree distribution as a strong indicator of GNN performance is novel. Furthermore, this indicator describes the dataset-level characteristics, allowing comparing GNN performance across different graph datasets. In addition, this paper presents a novel theoretical analysis, directly relating the degree distribution to the generalization performance of GNNs.

# 3 A Metadata-Driven Analysis on GNNs

## 3.1 Understanding GNNs with Metadata

**Motivation.** Real-world graph data are heterogeneous and incredibly diverse, contrasting with images or texts that often possess common structures or vocabularies. The inherent diversity of graph data makes it particularly challenging, if not unfeasible, to have one model to rule all tasks and datasets in the graph machine learning domain. Indeed, specific types of GNN models often only perform well on a selected set of graph learning datasets. For example, the expressive power of GNNs [41] is primarily relevant to graph-level prediction tasks rather than node-level tasks – higher-order GNNs with improved expressive power are predominantly evaluated on graph-level prediction tasks [30, 41]. As another example, several early GNNs such as Graph Convolution Networks (GCN) [19] or Graph Attention Networks (GAT) [37] only work well when the graphs exhibit homophily [45]. Consequently, it becomes crucial to identify and understand the critical data properties that influence the performance of different GNNs, allowing for more effective model design and selection.

The increasing availability of graph learning benchmarks that offer a wide range of structural and feature variations [16, 27] presents a valuable opportunity: one can possibly infer critical data properties from the performance of GNNs on these datasets. To systematically identify these critical data properties, we propose to conduct a regression analysis on the metadata of the benchmarks.

**Regression Analysis on Metadata.** In the regression analysis, the performance metrics of various GNN models on each dataset serve as the dependent variables, while the extracted data properties from each dataset act as the independent variables. Formally, we denote the number of datasets as $n$, the number of GNN models as $q$, and the number of data properties as $p$. Define the response variables $\{\mathbf{y}_i\}_{i \in [q]}$ to be GNN model performance operated on each dataset and the covariate variables $\{\mathbf{x}_j\}_{j \in [p]}$ to be properties of each dataset. Note that $\mathbf{y}_i \in \mathbb{R}^n, \forall i \in [q]$ and $\mathbf{x}_j \in \mathbb{R}^n, \forall j \in [p]$. For ease of notation, we define $\mathbf{Y} = (\mathbf{y}_1, ..., \mathbf{y}_q) \in \mathbb{R}^{n \times q}$ to be the response matrix of $n$ samples and $q$ variables, and $\mathbf{X} = (\mathbf{x}_1, ..., \mathbf{x}_p) \in \mathbb{R}^{n \times p}$ to be the covariate matrix of $n$ samples and $p$ variables.

Given these data matrices, we establish the following multivariate linear model to analyze the relationship between response matrix $\mathbf{Y}$ and covariate matrix $\mathbf{X}$, which is characterized by the coefficient matrix $\mathbf{B}$.

**Definition 3.1** (Multivariate Linear Model).

$$\mathbf{Y} = \mathbf{XB} + \mathbf{W}, \tag{1}$$

*where $\mathbf{B} \in \mathbb{R}^{p \times q}$ is the coefficient matrix and $\mathbf{W} = (\mathbf{w}_1, ..., \mathbf{w}_q) \in \mathbb{R}^{n \times q}$ is the matrix of error terms.*

Our goal is to find the most salient data properties that correlate with the performance of GNN models given a number of samples. To this end, we introduce two sparse regularizers for feature selections, which leads to the following Multivariate Sparse Group Lasso problem.

**Definition 3.2** (Multivariate Sparse Group Lasso Problem).

$$\underset{\mathbf{B}}{\arg\min} \frac{1}{2n} \|\mathbf{Y} - \mathbf{XB}\|_2^2 + \lambda_1 \|\mathbf{B}\|_1 + \lambda_g \|\mathbf{B}\|_{2,1}, \tag{2}$$

*where $\|\mathbf{B}\|_1 = \sum_{i=1}^p \sum_{j=1}^q |\mathbf{B}_{ij}|$ is the $L_1$ norm of $\mathbf{B}$, $\|\mathbf{B}\|_{2,1} = \sum_{i=1}^p \sqrt{\sum_{j=1}^q \mathbf{B}_{ij}^2}$ is the $L_{2,1}$ group norm of $\mathbf{B}$, and $\lambda_1, \lambda_g > 0$ are the corresponding penalty parameters.*

In particular, the $L_1$ penalty encourages the coefficient matrix $\mathbf{B}$ to be sparse, only selecting salient data properties. The $L_{2,1}$ penalty further leverages the structure of the dependent variables and tries to make only a small set of the GNN models' performance depends on each data property, thus differentiating the impacts on different GNNs.

To solve for the coefficient matrix $\mathbf{B}$ in Equation 2, we employ an R package, MSGLasso [21], using matrices $\mathbf{Y}$ and $\mathbf{X}$ as input. To ensure proper input for the MSGLasso solver [21], we have preprocessed the data by standardizing the columns of both $\mathbf{Y}$ and $\mathbf{X}$.

## 3.2 Data Properties and Model Performance

Next, we introduce the metadata used for the regression analysis. We obtain both the benchmark datasets and the model performance using the Graph Learning Indexer (GLI) library [27].

**Data Properties.**  We include the following benchmark datasets in our regression analysis: cora [42], citeseer [42], pubmed [42], texas [33], cornell [33], wisconsin [33], actor [33], squirrel [33], chameleon [33], arxiv-year [23], snap-patents [23], penn94 [23], pokec [23], genius [23], and twitch-gamers [23]. For each graph dataset, we calculate 15 data properties, which can be categorized into the following six groups:

- *Basic*: Edge Density, Average Degree, Degree Assortativity;

- *Distance*: Pseudo Diameter;

- *Connectivity*: Relative Size of Largest Connected Component (RSLCC);

- *Clustering*: Average Clustering Coefficient (ACC), Transitivity, Degeneracy;

- *Degree Distribution*: Gini Coefficient of Degree Distribution (Gini-Degree);

- *Attribute*: Edge Homogeneity, In-Feature Similarity, Out-Feature Similarity, Feature Angular SNR, Homophily Measure, Attribute Assortativity.

The formal definition of these graph properties can be found in Appendix A.

**Model Performance.**  For GNN models, we include GCN [19], GAT [37], GraphSAGE [13], MoNet [29], MixHop [1], and LINKX [23] into our regression analysis. We also include a non-graph model, Multi-Layer Perceptron (MLP). The complete experimental setup for these models can be found in Appendix B.

Table 1: The estimated coefficient matrix **B** of the multivariate sparse regression analysis. Each entry indicates the strength (magnitude) and direction $(+, -)$ of the relationship between a graph data property and the performance of a GNN model. The six most salient data properties are indicated in **bold**.

| Graph Data Property | GCN | GAT | GraphSAGE | MoNet | MixHop | LINKX | MLP |
|---|---|---|---|---|---|---|---|
| Edge Density | 0 | 0 | 0 | 0 | 0 | 0.0253 | 0.0983 |
| **Average Degree** | 0.2071 | 0 | 0.1048 | 0.1081 | 0 | 0.3363 | 0 |
| **Pseudo Diameter** | 0 | -0.349 | -0.1531 | 0 | -0.4894 | -0.3943 | -0.6119 |
| Degree Assortativity | 0 | 0 | 0 | -0.0744 | 0 | 0 | 0 |
| RSLCC | 0.1019 | 0 | 0 | 0.0654 | 0 | 0.1309 | 0 |
| ACC | 0 | 0 | 0 | 0 | 0 | 0 | -0.0502 |
| Transitivity | 0 | -0.0518 | 0 | -0.1372 | 0 | 0.2311 | 0 |
| Degeneracy | 0 | 0 | 0 | 0 | 0 | 0 | -0.1657 |
| **Gini-Degree** | -0.4403 | -0.2961 | -0.3267 | -0.2944 | -0.4205 | -0.367 | -0.1958 |
| **Edge Homogeneity** | 0.7094 | 0.4705 | 0.7361 | 0.8122 | 0.6407 | 0.2006 | 0.4776 |
| **In-Feature Similarity** | 0.3053 | 0.1081 | 0.1844 | 0.1003 | 0.4613 | 0.6396 | 0.2399 |
| Out-Feature Similarity | 0 | 0 | 0 | 0 | 0 | 0 | 0 |
| **Feature Angular SNR** | 0.2522 | 0 | 0.2506 | 0 | 0.2381 | 0.3563 | 0.3731 |
| Homophily Measure | 0 | 0.4072 | 0 | 0 | 0 | 0 | 0 |
| Attribute Assortativity | 0 | 0 | 0 | 0 | 0 | 0 | 0 |

## 3.3 Analysis Results

The estimated coefficient matrix **B** is presented in Table 1. As can be seen, the estimated coefficient matrix is fairly sparse, allowing us to identify salient data properties. Next, we will discuss the six most salient data properties that correlate to some or all of the GNN models' performance. For the data properties that have an impact on all GNNs' performance, we call them **Widely Influential Factors**; for the data properties that have an impact on over one-half of GNNs' performance, we call them **Narrowly Influential Factors**. Notice that the $(+, -)$ sign after the name of the factors indicates whether this data property has a positive or negative correlation with the GNN performance.

**Widely Influential Factors.** We discover that the Gini coefficient of the degree distribution (Gini-Degree), Edge Homogeneity, and In-Feature Similarity impact all GNNs' model performance consistently.

- *Gini-Degree* $(-)$ measures how the graph's degree distribution deviates from the perfectly equal distribution, i.e., a regular graph. This is a crucial data property that dramatically influences GNNs' performance but remains under-explored in prior literature.

- *Edge Homogeneity* $(+)$ is a salient indicator for all GNN models' performance. This phenomenon coincides with the fact that various GNNs assume strong homophily condition [28] to obtain improvements on node classification tasks [13, 19, 37].

- *In-feature Similarity* $(+)$ calculates the average of feature similarity within each class. Under the homophily assumption, GNNs work better when nodes with the same labels additionally have similar node features, which also aligns with existing findings in the literature [15].

**Narrowly Influential Factors.** We find that Average Degree, Pseudo Diameter, and Feature Angular SNR are salient factors for a subset of GNN models, although we do not yet have a good understanding on the mechanism of how these data properties impact model performance.

- *Average Degree* $(+)$ is more significant for GCN, GraphSAGE, MoNet, and LINKX.

- *Pseudo Diameter* $(-)$ is more significant for GAT, GraphSAGE, MixHop, LINKX, and MLP.

- *Feature Angular SNR* $(+)$ is more significant for GCN, GraphSAGE, MixHop, LINKX, and MLP.

We note that the regression analysis only indicates associative relationships between data properties and the model performance. While our analysis has successfully identified well-known influential

data properties, e.g., Edge Homogeneity, the mechanism for most identified data properties through which they impact the GNN performance remains under-explored.

To further verify the effectiveness of the proposed metadata-driven approach in identifying critical data properties, we perform an in-depth analysis for *Gini-Degree*, one of the most widely influential factors. In the following Section 4 and 5, we conduct theoretical analysis and controlled experiments to understand how Gini-Degree influences GNNs' performance.

# 4    Theoretical Analysis on the Impact of Degree Distribution

In this section, we present a theoretical analysis on influence of graph data's degree distribution on the performance of GNNs. Specifically, our analysis investigates the linear separability of node representations produced by applying graph convolution to the node features. In the case that the graph data comes from a Degree-Corrected Stochastic Block Model, we show that nodes from different classes are more separable when their degree exceeds a threshold. This separability result relates the graph data's degree distribution to the GNN performance. Finally, we discuss the role of Gini-Degree on the GNN performance using implications of our theory.

## 4.1    Notations and Sketch of Analysis

**The Graph Data.**    Let $\mathcal{G} = \{\mathcal{V}, \mathcal{E}\}$ be an undirected graph, where $\mathcal{V}$ is the set of nodes and $\mathcal{E}$ is the set of edges. The information regarding the connections within the graph can also be summarized as an adjacency matrix $\mathbf{A} \in \{0, 1\}^{|\mathcal{V}| \times |\mathcal{V}|}$, where $|\mathcal{V}|$ is the number of nodes in the graph $\mathcal{G}$. Each node $i \in \mathcal{V}$ possesses a $d$-dimensional feature vector $\mathbf{x}_i \in \mathbb{R}^d$. The features for all nodes in $\mathcal{G}$ can be stacked and represented as a feature matrix $\mathbf{X} \in \mathbb{R}^{|\mathcal{V}| \times d}$. In the context of node classification, each node $i$ is associated with a class label $y_i \in \mathcal{C}$, where $\mathcal{C}$ is the set of labels.

**Graph Convolutional Network [19].**    In our analysis, we consider a single-layer graph convolution, which can be defined as an operation on the adjacency matrix and feature matrix of a graph $\mathcal{G}$ to produce a new feature matrix $\tilde{\mathbf{X}}$. Formally, the output of a single-layer graph convolution operation can be represented as $\tilde{\mathbf{X}} = \mathbf{D}^{-1}\tilde{\mathbf{A}}\mathbf{X}$, where $\tilde{\mathbf{A}} = \mathbf{A} + \mathbf{I}$ is the augmented adjacency matrix with added self-loops, and $\mathbf{D}$ is the diagonal degree matrix with $\mathbf{D}_{ii} = \deg(i) = \sum_{j \in [n]} \tilde{\mathbf{A}}_{ij}$. Hence, for each node $i \in \mathcal{V}$, the new node representation will become $\tilde{\mathbf{x}}_i \in \mathbb{R}^d$, which is the $i$th row of the output matrix $\tilde{\mathbf{X}}$.

**Sketch of Our Analysis.**    Our analysis builds upon and generalizes the theoretical framework introduced by Baranwal et al. [3], where they demonstrate that in comparison to raw node features, the graph convolution representations of nodes have better linear separability if the graph data comes from Contextual Stochastic Block Model (CSBM) [4, 8]. However, in CSBM, the nodes within the same class all have similar degrees with high probability, which prevents us to draw meaningful conclusions about the impact of degree distribution.

To better understand the role of degree distribution in the GNN performance, we develop a non-trivial generalization of the theory by Baranwal et al. [3]. Specifically, we first coin a new graph data generation model, Degree-Corrected Contextual Stochastic Block Model (DC-CSBM) that combines and generalizes Degree-Corrected SBM (DC-SBM) [17] and CSBM, and leverages heterogeneity in node degrees into consideration. Under DC-CSBM, we find that node degrees play a crucial role in the statistical properties of the node representations, and the node degrees have to exceed a certain threshold in order for the node representations to sufficiently leverage the neighborhood information and become reliably separable. Notably, the incorporation of the node degree heterogeneity into the analysis requires a non-trivial adaptation of the analysis by Baranwal et al. [3].

## 4.2    Degree-Corrected Contextual Stochastic Block Model (DC-CSBM)

In this section, we introduce the DC-CSBM that models the generation of graph data. Specifically, we assume the graph data is randomly sampled from a DC-CSBM with 2 classes.

**DC-CSBM With 2 Classes.** Let us define the class assignments $(\epsilon_i)_{i \in [n]}$ as independent and identically distributed (i.i.d.) Bernoulli random variables coming from $\mathrm{Ber}(\frac{1}{2})$, where $n = |\mathcal{V}|$ is the number of nodes in the graph $\mathcal{G}$. These class assignments divide $n$ nodes into 2 classes: $C_0 = \{i \in [n] : \epsilon_i = 0\}$ and $C_1 = \{i \in [n] : \epsilon_i = 1\}$. Assume that inter-class edge probability is $q$ and intra-class edge probability is $p$, and no self-loops are allowed. For each node $i$, we additionally introduce a degree-correction parameter $\theta_i \in (0, n]$, which can be interpreted as the propensity of node $i$ to connect with others. Note that to keep the DC-SBM identifiable and easier to analyze, we adopt a normalization rule to enforce the following constraint: $\sum_{i \in C_0} \theta_i = |C_0|$, $\sum_{i \in C_1} \theta_i = |C_1|$ and thus $\sum_{i \in \mathcal{V}} \theta_i = n$.

**Assumptions on Adjacency Matrix and Feature Matrix.** Conditioning on $(\epsilon_i)_{i \in [n]}$, each entry of the adjacency matrix $\mathbf{A}$ is a Poisson random variable with $\mathbf{A}_{ij} \sim \mathrm{Poi}(\theta_i \theta_j p)$ if $i, j$ are in the same class and $\mathbf{A}_{ij} \sim \mathrm{Poi}(\theta_i \theta_j q)$ if $i, j$ are in different classes. On top of this, let $\mathbf{X} \in \mathbb{R}^{n \times d}$ be the feature matrix where each row $\mathbf{x}_i$ represents the node feature of node $i$. Assume each $\mathbf{x}_i$ is an independent $d$-dimensional Gaussian random vector with $\mathbf{x}_i \sim \mathcal{N}(\boldsymbol{\mu}, \frac{1}{d}\mathbf{I})$ if $i \in C_0$ and $\mathbf{x}_i \sim \mathcal{N}(\boldsymbol{\nu}, \frac{1}{d}\mathbf{I})$ if $i \in C_1$. We let $\boldsymbol{\mu}, \boldsymbol{\nu} \in \mathbb{R}^d$ to be fixed $d$-dimensional vectors with $\|\boldsymbol{\mu}\|_2, \|\boldsymbol{\nu}\|_2 \leq 1$, which serve as the Gaussian mean for the two classes.

Given a particular choice of $n, \boldsymbol{\mu}, \boldsymbol{\nu}, p, q$ and $\theta = (\theta_i)_{i \in [n]}$, we can define a class of random graphs generated by these parameters and sample a graph from such DC-CSBM as $\mathcal{G} = (\mathbf{A}, \mathbf{X}) \sim$ DC-CSBM$(n, \boldsymbol{\mu}, \boldsymbol{\nu}, p, q, \theta)$.

### 4.3 Linear Separability After Graph Convolution

**Linear Separability.** Linear separability refers to the ability to linearly differentiate nodes in the two classes based on their feature vectors. Formally, for any $\mathcal{V}_s \subseteq \mathcal{V}$, we say that $\{\tilde{\mathbf{x}}_i : i \in \mathcal{V}_s\}$ is linearly separable if there exists some unit vector $\mathbf{v} \in \mathbb{R}^d$ and a scalar $b$ such that $\mathbf{v}^\top \tilde{\mathbf{x}}_i + b < 0, \forall i \in C_0 \cap \mathcal{V}_s$ and $\mathbf{v}^\top \tilde{\mathbf{x}}_i + b > 0, \forall i \in C_1 \cap \mathcal{V}_s$. Note that linear separability is closely related to GNN performance. Intuitively, more nodes being linearly separable will lead to better GNN performance.

**Degree-Thresholded Subgroups of $C_0$ and $C_1$.** To better control the behavior of graph convolution operation, we will focus on particular subgroups of $C_0$ and $C_1$ where the member nodes having degree-corrected factor larger or equal to a pre-defined threshold $\alpha > 0$. Slightly abusing the notations, we denote these subgroups as $C_0(\alpha)$ and $C_1(\alpha)$, which are formally defined below.

**Definition 4.1** ($\alpha$-Subgroups). *Given any $\alpha \in (0, n]$, define $\alpha$-subgroups of $C_0$ and $C_1$ as follows:*

$$C_0(\alpha) = \{j \in [n] : \theta_j \geq \alpha \text{ and } j \in C_0\},$$
$$C_1(\alpha) = \{j \in [n] : \theta_j \geq \alpha \text{ and } j \in C_1\}.$$

Let $\mathcal{V}_\alpha := C_0(\alpha) \cup C_1(\alpha)$, we are interested in analyzing the linear separability of the node representations after the graph convolution operation, namely $\{\tilde{\mathbf{x}}_i : i \in \mathcal{V}_\alpha\}$. Recall that for each node $i$, $\tilde{\mathbf{x}}_i = \frac{1}{\deg(i)} \sum_{j \in \mathcal{N}(i)} \mathbf{x}_j$, where $\mathcal{N}(i)$ is the set of neighbors of node $i$.

**Relationship Between $\alpha$ and Linear Separability.** We first make the following assumptions about the DC-CSBM, closely following the assumptions made by Baranwal et al. [3].

**Assumption 4.2** (Graph Size). *Assume the relationship between the graph size $n$ and the feature dimension $d$ follows $\omega(d \log d) \leq n \leq O(poly(d))$.*

**Assumption 4.3** (Edge Probabilities). *Define $\Gamma(p, q) := \frac{p-q}{p+q}$. Assume the edge probabilities $p, q$ satisfy $p, q = \omega(\log^2(n)/n)$ and $\Gamma(p, q) = \Omega(1)$.*

Theorem 4.4 asserts that if the threshold $\alpha$ is not too small, then the set $\mathcal{V}_\alpha = C_0(\alpha) \cup C_1(\alpha)$ can be linear separated with high probability. The proof of Theorem 4.4 can be found in Appendix C.

**Theorem 4.4** (Linear Separability of $\alpha$-Subgroups). *Suppose that Assumption 4.2 and 4.3 hold. For any $(\mathbf{X}, \mathbf{A}) \sim$ DC-CSBM$(n, \boldsymbol{\mu}, \boldsymbol{\nu}, p, q, \theta)$, if $\alpha = \omega\left(\max\left(\frac{1}{\log n}, \frac{\log n}{dn(p+q)\|\boldsymbol{\mu}-\boldsymbol{\nu}\|_2^2}\right)\right)$, then*

$$\mathbb{P}(\{\tilde{\mathbf{x}}_i : i \in \mathcal{V}_\alpha\} \text{ is linearly separable}) = 1 - o_d(1),$$

*where $o_d(1)$ is a quantity that converges to 0 as $d$ approaches infinity.*

Note that Theorem 4.4 suggests that, when the heterogeneity of node degrees is taken into consideration, the nodes with degrees exceeding a threshold $\alpha$ are more likely to be linearly separable. And the requirement for the threshold $\alpha$ depends on the DC-CSBM parameters: $n, p, q, \boldsymbol{\mu}, \boldsymbol{\nu}$.

**Remark 4.5.** *If we let $p, q \in \Theta(\frac{\log^3 n}{n})$ and $\|\boldsymbol{\mu} - \boldsymbol{\nu}\|_2$ be fixed constant, then the requirement can be reduced to $\alpha \in \omega(\frac{1}{\log n})$, which is not too large. Given this particular setting and reasonable selection of $p, q$, the regime of acceptable $\alpha$ is broad and thus demonstrates the generalizability of Theorem 4.4.*

### 4.4  Implications on Gini-Degree

Finally, we qualitatively discuss the relationship between Gini-Degree and GNNs' performance using the results from Theorem 4.4. For any $\alpha > 0$ that meets the criteria in the statement, we can consider,

1. *Negative correlation between Gini-Degree and the size of $\mathcal{V}_\alpha$*: If the number of nodes and edges is fixed, a higher Gini-Degree implies more high-degree nodes in the network and thus the majority of nodes are receiving lower degrees. Clearly, if most of the nodes have lower degrees, then there will be fewer nodes having degrees exceeding a certain threshold proportional to $\alpha^1$ and being placed in $\mathcal{V}_\alpha$. Hence, a dataset with a higher (or lower) Gini-Degree will lead to a smaller (or larger) size of $\mathcal{V}_\alpha$.

2. *Positive correlation between the size of $\mathcal{V}_\alpha$ and model performance*: Intuitively, the GNN performance tends to be better if there are more nodes that can be linearly separable after graph convolution. Consequently, the GNN performance is positively relevant to the size of $\mathcal{V}_\alpha$ corresponding to the minimum possible $\alpha$.

Combining the two factors above, our analysis suggests that Gini-Degree tends to have a negative correlation with GNNs' performance.

## 5  Controlled Experiment on Gini-Degree

To further verify whether there is a causal relationship between the degree distribution of graph data (in particular, measured by Gini-Degree) and the GNN performance, we conduct a controlled experiment using synthetic graph datasets.

**Experiment Setup.** We first generate a series of synthetic graph datasets using the GraphWorld library [31]. To investigate the causal effect of Gini-Degree, we manipulate the data generation parameters to obtain datasets with varying Gini-Degree while keeping a bunch of other properties fixed. Specifically, we use the SBM generator in GraphWorld library and set the number of nodes $n = 5000$, the average degree as 30, the number of clusters as 4, cluster size slope as 0.5, feature center distance as 0.5, the edge probability ratio $p/q = 4.0$, feature dimension as 16, feature cluster variance as 0.05. The parameters above are fixed throughout our experiments, and their complete definition can be found in the Appendix. By manipulating the power law exponent parameter of the generator, we obtain five synthetic datasets with Gini-Degree as $0.906, 0.761, 0.526, 0.354$, and $0.075$, respectively.

Then we train the same set of GNN models and MLP model as mentioned in Table 1 on each dataset. We randomly split the nodes into training, validation, and test sets with a ratio of 3:1:1. We closely follow the hyperparameters and the training protocol in the GLI library [27], which is where we obtain the metadata in Section 3. We run five independent trials with different random seeds.

**Experiment Results.** The experiment results are shown in Table 2. We observe an evident monotonically decreasing trend for the performance of the graph-based models, GCN, GAT, GraphSAGE, MoNet, MixHop, and LINKX, as Gini-Degree increases. However, there is no clear pattern for the non-graph model, MLP. This result suggests that these widely-used GNN models are indeed sensitive to Gini-Degree, which validates our result of sparse regression analysis. Note that MLP does not take the graph structure into consideration, and hence the degree distribution has less influence

---

[1]Note that the expected value of the degree of node $i$ is proportional to $\theta_i$ when we ignore self-loops. (See Appendix C for more information.) Thus, the lower bound $\alpha$ on degree-corrected factors can be translated to the lower bound $n(p + q)\alpha$ on degrees.

Table 2: Controlled experiment results for varying *Gini-Degree*. Standard deviations are derived from 5 independent runs. The performances of all models except for MLP have an evident negative correlation with *Gini-Degree*.

| Gini-Degree | GCN | GAT | GraphSAGE | MoNet | MixHop | LINKX | MLP |
|---|---|---|---|---|---|---|---|
| 0.906 | 0.798±0.004 | 0.659±0.01 | 0.76±0.005 | 0.672±0.002 | 0.804±0.005 | 0.832±0.002 | 0.595±0.006 |
| 0.761 | 0.817±0.001 | 0.732±0.005 | 0.818±0.004 | 0.696±0.015 | 0.817±0.004 | 0.849±0.002 | 0.756±0.002 |
| 0.526 | 0.874±0.004 | 0.742±0.006 | 0.825±0.013 | 0.8±0.028 | 0.826±0.003 | 0.853±0.002 | 0.655±0.005 |
| 0.354 | 0.906±0.002 | 0.737±0.008 | 0.857±0.008 | 0.83±0.013 | 0.837±0.002 | 0.867±0.002 | 0.66±0.07 |
| 0.075 | 0.948±0.002 | 0.746±0.005 | 0.878±0.002 | 0.92±0.002 | 0.84±0.002 | 0.893±0.001 | 0.705±0.002 |

on the performance of MLP. The result on MLP also indicates that we have done a reasonably well-controlled experiment.

## 6 Conclusion

In this work, we propose a novel metadata-driven approach that can efficiently identify critical graph data properties influencing the performance of GNNs. This is a significant contribution given the diverse nature of graph-structured data and the sensitivity of GNN performance to these specific properties. We also verify the effectiveness of the proposed approach through an in-depth case study around one identified salient graph data property.

As a side product, this paper also highlights the considerable impact of the degree distribution, a salient data property identified through our metadata-driven regression analysis, on the GNN performance. We present a novel theoretical analysis and a carefully controlled experiment to demonstrate this impact.

**Acknowledgement**

The authors would like to thank Pingbang Hu for the feedback on the draft.

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

# A    Definitions of Dataset Properties

We introduce the formal definitions of the dataset properties mentioned in Section 3.2. Following the definitions in Section 4.1, we further define $n = |\mathcal{V}|$ and $m = |\mathcal{E}|$ to denote the number of nodes and edges of graph $\mathcal{G}$. Also, in the context of the node classification task, we define $\mathcal{Y} \in \mathbb{R}^n$ as the vector of node labels and $C$ as the number of classes.

## A.1    Basic

**Edge Density:** The edge density for an undirected graph is calculated as $\frac{2m}{n(n-1)}$, while for a directed graph, it is computed as $\frac{m}{n(n-1)}$.

**Average Degree:** The average degree for an undirected graph is defined as $\frac{2m}{n}$, while for a directed graph, it is defined as $\frac{m}{n}$.

**Degree Assortativity:** The degree assortativity is the average Pearson correlation coefficient of all pairs of connected nodes. It quantifies the tendency of nodes in a network to be connected to nodes with similar or dissimilar degrees and ranges between -1 and 1.

## A.2    Distance

**Pseudo Diameter:** The pseudo diameter is an approximation of the diameter of a graph and provides a lower bound estimation of its exact value.

## A.3    Connectivity

**Relative Size of Largest Connected Component (RSLCC):** The relative size of the largest connected component is determined by calculating the ratio between the size of the largest connected component and $n$.

## A.4    Clustering

**Average Clustering Coefficient (ACC):** First define $T(u)$ as the number of triangles including node $u$, then the local clustering coefficient for node $u$ is calculated as $\frac{2}{\deg(u)(\deg(u)-1)}T(u)$ for undirected graph, where $\deg(u)$ is the degree of node $u$; and is calculated as $\frac{2}{\deg^{tot}(u)(\deg^{tot}(u)-1)-2\deg^{\leftrightarrow}(u)}T(u)$ for directed graph, where $\deg^{tot}(u)$ is the sum of in-degree and out-degree of node $u$ and $\deg^{\leftrightarrow}(u)$ is the reciprocal degree of $u$. The average clustering coefficient is then defined as the average local clustering coefficient of all the nodes in the graph.

**Transitivity:** The transitivity is defined as the fraction of all possible triangles present in the graph. Formally, it can be written as $3\frac{\#\text{triangles}}{\#\text{triads}}$, where a triad is a pair of two edges with a shared vertex.

**Degeneracy:** The degeneracy is determined as the least integer $k$ such that every induced subgraph of the graph contains a vertex with its degree smaller or equal to $k$.

## A.5    Degree Distribution

**Gini Coefficient of Degree Distribution (Gini-Degree):** The Gini coefficient of the node degrees of the graph.

## A.6    Attribute

**Edge Homogeneity [31]:** The edge homogeneity is defined as the ratio of edges whose endpoints have the same node labels.

**In-Feature Similarity [31]:** First define within-class angular feature similarity as $1-$ angular_distance$(\mathbf{x}_i, \mathbf{x}_j)$ for an edge $(i, j)$ with its endpoints $i$ and $j$ have the same node labels. In-Feature Similarity is the average within-class angular feature similarity of all such edges in the graph.

**Out-Feature Similarity [31]:** First define between-class angular feature similarity as $1-$ angular_distance$(\mathbf{x}_i, \mathbf{x}_j)$ for an edge $(i,j)$ with its endpoints $i$ and $j$ have different node labels. Out-Feature Similarity is the average between-class angular feature similarity of all such edges in the graph.

**Feature Angular SNR [31]:** The feature angular SNR is computed as the ratio between in-feature similarity and out-feature similarity.

**Homophily Measure [23]:** The homophily measure is defined as

$$\hat{h} = \frac{1}{C-1}\sum_{k=1}^{C}[h_k - \frac{|C_k|}{n}]_+,$$

where $[a]_+ = \max(a,0)$, $|C_k|$ is the total number of nodes having their label $k$ and $h_k$ is the class-wise homophily metric defined below,

$$h_k = \frac{\sum_{u:\mathcal{Y}_u=k} d_u^{(\mathcal{Y}_u)}}{\sum_{u:\mathcal{Y}_u=k} d_u},$$

where $d_u$ is the number of neighbors of node $u$ and $d_u^{(\mathcal{Y}_u)}$ is the number of neighbors of node $u$ having the same node label.

**Attribute Assortativity:** The attribute assortativity is the average Pearson correlation coefficient of all pairs of connected nodes. It quantifies the tendency of nodes in a network to be connected to nodes with the same or different attributes (here node label) and ranges between -1 and 1.

## B    Experiment Setup for Obtaining Metadata

In this section, we describe more details of the experimental setup to obtain GNNs' performance that we use in Section 3.2, mostly following Ma et al. [27]. For completeness, we list down the model setting used by them in the following paragraphs.

GCN [19], GAT [37], GraphSAGE [13], MoNet [29], MLP, and MixHop [1] are set to have two layers with hidden dimension equals to 8. For LINKX [23], $MLP_A$, $MLP_X$ are set to be a one-layer network and $MLP_f$ to be a two-layers network, following the setting in Lim et al. [23].

For the rest of the training settings, we adopt the same configuration for all experiments. Specifically, we set learning rate = 0.01, weight decay = 0.001, dropout rate = 0.6, max epoch = 10000, and batch size = 256. We use Adam [18] as an optimizer for all models except LINKX. AdamW [26] is used with LINKX in order to comply with Lim et al. [23]. For datasets with binary labels (i.e., penn94, pokec, genius, and twitch-gamers), we choose the ROC AUC score as the evaluation metric; while for other datasets, we use test accuracy instead.

We also let all the detailed model settings remain consistent with the same with Ma et al. [27]. Namely,

- GAT: Number of heads in multi-head attention = 8. LeakyReLU angle of negative slope = 0.2. No residual is applied. The dropout rate on attention weight is the same as the overall dropout.
- GraphSAGE: Aggregator type is GCN. No norm is applied.
- MoNet: Number of kernels = 3. Dimension of pseudo-coordinte = 2. Aggregator type = sum.
- MixHop: List of powers of adjacency matrix = $[1,2,3]$. No norm is applied. Layer Dropout rate = 0.9.
- LINKX: No inner activation.

## C    Proof of Theorem 4.4

**Proof Sketch.**    To prove Theorem 4.4, we first show that degree and the neighborhood distribution of each node concentrate with high probability. Then we claim that the node features after the

convolution operation will be centered around specific mean values, depending on the node classes. Finally, we demonstrate that the nodes in different classes can be linearly separated by the hyperplane passing through the mid-point of the two mean values $\mu, \nu$ with high probability.

We prove the intermediate results in Lemma C.4 (degree and neighborhood distribution concentration inequalities) by utilizing Lemma C.5 (Chernoff bound for Poisson random variable) and in Lemma C.6 (convoluted feature concentration) by making use of Lemma C.7 (Borell's inequality). Finally, given the requirement of $\alpha$ stated in Theorem 4.4, we argue that the convoluted node features in two classes can be linearly separated with a high level of confidence.

**Novelty of our Proof.** The general structure of our proof follows that of Baranwal et al. [3]. However, our analysis requires non-trivial adaptation of the proof by Baranwal et al. [3]. This is because we have a more general data model, DC-CSBM, where the CSBM assumed by Baranwal et al. [3] is a restricted special case of ours. In particular, we assume each edge is generated by the Poisson random variable following DC-SBM, instead of the Bernoulli random variable assumed by CSBM; we also incorporate the degree-corrected factor in our analysis to model node degree heterogeneity within communities, which gives us the flexibility to discuss linear separability for subgraphs with different levels of sparsity.

Before we state and prove Lemma C.4, let us first define the following events that we will work on.

**Definition C.1** (Class Size Concentration). *For any $\delta > 0$, define*

$$\mathbf{I}_1(\delta) = \left\{ \frac{n}{2}(1 - \delta) \leq |C_0|, |C_1| \leq \frac{n}{2}(1 + \delta) \right\}.$$

**Definition C.2** (Degree Concentration). *For any $\delta' > 0$ and for each node $i \in [n]$, define*

$$\mathbf{I}_{2,i}(\delta') = \left\{ \frac{1}{2}(p + q)(1 - \delta')\theta_i \leq \frac{D_{ii}}{n} \leq \frac{1}{2}(p + q)(1 + \delta')\theta_i \right\}.$$

**Definition C.3** (Neighborhood Distribution Concentration). *For any $\delta' > 0$ and for each node $i \in [n]$, define*

$$\mathbf{I}_{3,i}(\delta') = \left\{ \frac{(1 - \epsilon_i)p + \epsilon_i q}{p + q}(1 - \delta') \leq \frac{|C_0 \cap \mathcal{N}_i|}{D_{ii}} \leq \frac{(1 - \epsilon_i)p + \epsilon_i q}{p + q}(1 + \delta') \right\}$$
$$\bigcap \left\{ \frac{(1 - \epsilon_i)q + \epsilon_i p}{p + q}(1 - \delta') \leq \frac{|C_1 \cap \mathcal{N}_i|}{D_{ii}} \leq \frac{(1 - \epsilon_i)q + \epsilon_i p}{p + q}(1 + \delta') \right\},$$

*where $\mathcal{N}_i$ denotes the set of nodes connected to node $i$.*

Then in Lemma C.4, we argue that for nodes in the $\alpha$-subgroup defined in 4.1 for some appropriately chosen $\alpha > 0$, the above events will happen simultaneously with high probability.

**Lemma C.4** (Concentration Inequalities). *Given $\alpha \in (\frac{1}{\log n}, n]$, $C_0(\alpha), C_1(\alpha)$ defined by Definition 4.1, and $\mathcal{V}_\alpha = C_0(\alpha) \cup C_1(\alpha)$. Let $\delta = n^{-1/2+\epsilon}$ and $\delta' = (\alpha \log n)^{-1/2+\epsilon}$, then for $\epsilon > 0$ small enough, we have for any $c > 0$, there is some $C > 0$ such that*

$$\mathbb{P}\left( \mathbf{I}_1(\delta) \bigcap_{i \in \mathcal{V}_\alpha} \mathbf{I}_{2,i}(\delta') \bigcap_{i \in \mathcal{V}_\alpha} \mathbf{I}_{3,i}(\delta') \right) \geq 1 - \frac{C}{n^c}.$$

*Proof.* Firstly, we consider the event $\mathbf{I}_1(\delta)$. Since $(\epsilon_i)_{i \in [n]} \sim \text{Ber}(\frac{1}{2})$, by the Hoeffding's inequality for independent Bernoulli random variables [38, Theorem 2.2.6], we have for any $\delta > 0$ that

$$\mathbb{P}\left( \left| \frac{1}{n} \sum_{i=1}^{n} \epsilon_i - \frac{1}{2} \right| \geq \delta/2 \right) \leq 2 \exp(-n\delta^2/2).$$

Notice that $\sum_{i=1}^{n} \epsilon_i = |C_1|$ and $|C_0| + |C_1| = n$, we can conclude that for any $\delta > 0$, the probability that the number of nodes in each class concentrates will satisfy

$$\mathbb{P}\left( \frac{|C_0|}{n}, \frac{|C_1|}{n} \in \left[ \frac{1}{2} - \delta, \frac{1}{2} + \delta \right] \right) \geq 1 - C \exp(-cn\delta^2),$$

for some constant $C, c > 0$.

We now turn to the events $\{\mathbf{I}_{2,i}(\delta')\}_{i \in [n]}$. Notice that the node degrees are sums of independent Poisson random variables. It is known that sums of independent Poisson random variables will be another Poisson random variable. Hence, conditioning on $\theta = (\theta_i)_{i \in [n]}$, for each node $i \in [n]$, we have

$$D_{ii} \sim 1 + \mathrm{Poi}(\frac{n-1}{2}(p+q)\theta_i),$$

where $D_{ii}$ is the degree of node $i$, and

$$\mathbb{E}[D_{ii}] = 1 + \frac{n-1}{2}(p+q)\theta_i.$$

To prove that $\mathbf{I}_{2,i}(\delta')$ will occur with high probability for each $i \in [n]$, we introduce the following result [38, Corollary 2.3.7] whose proof can be found in the referred literature:

**Lemma C.5** (Corollary 2.3.7 [38]). *If $X \sim Poi(\lambda)$, then for $t \in (0, \lambda]$, we have*

$$\mathbb{P}(|X - \lambda| \geq t) \leq 2 \exp(-\frac{ct^2}{\lambda}).$$

Here, we can let $t = \delta'\lambda$ where $\delta' \in (0, 1]$ and get a tail bound as follows:

$$\mathbb{P}(|D_{ii} - \mathbb{E}[D_{ii}]| \geq \delta'\mathbb{E}[D_{ii}]) \leq 2 \exp(-\frac{c(\delta'\mathbb{E}[D_{ii}])^2}{\mathbb{E}[D_{ii}]}) = 2 \exp(-c\mathbb{E}[D_{ii}]\delta'^2).$$

It follows that for each $i \in [n]$ and any $\delta' \in (0, 1]$, we have

$$\mathbb{P}\left(\frac{D_{ii}}{n} \in \left[\frac{1}{2}(p+q)(1-\delta')\theta_i, \frac{1}{2}(p+q)(1+\delta')\theta_i\right]^c\right) \leq C \exp(-cn(p+q)\theta_i\delta'^2),$$

for some $C, c > 0$.

We next consider the events $\{\mathbf{I}_{3,i}(\delta')\}_{i \in [n]}$. Observe that for each node $i$, we can decompose node degree as $D_{ii} = D_{ii}^{\mathrm{intra}} + D_{ii}^{\mathrm{inter}}$, where

$$D_{ii}^{\mathrm{intra}} = \sum_{j \in \mathcal{N}(i)} \mathbb{1}\{\epsilon_j = \epsilon_i\},$$

and

$$D_{ii}^{\mathrm{inter}} = \sum_{j \in \mathcal{N}(i)} \mathbb{1}\{\epsilon_j \neq \epsilon_i\}.$$

Obviously, $D_{ii}^{\mathrm{intra}} = |C_{\epsilon_i} \cap \mathcal{N}_i|$ and $D_{ii}^{\mathrm{inter}} = |C_{1-\epsilon_i} \cap \mathcal{N}_i|$ will concentrate around $\frac{np\theta_i}{2}$ and $\frac{nq\theta_i}{2}$, correspondingly. And given the tail bound for $\{\mathbf{I}_{2,i}(\delta')\}_{i \in [n]}$, by a similar argument, we have for each $i \in [n]$ and any $\delta' \in (0, 1]$,

$$\mathbb{P}(\mathbf{I}_{3,i}(\delta')) \geq 1 - C \exp(-cn(p+q)\theta_i\delta'^2),$$

for some $C, c > 0$.

Define the union event $U(\delta, \delta') = \mathbf{I}_1(\delta) \bigcap_{i \in \mathcal{V}_\alpha} \mathbf{I}_{2,i}(\delta') \bigcap_{i \in \mathcal{V}_\alpha} \mathbf{I}_{3,i}(\delta')$. Recall that $\forall i \in \mathcal{V}_\alpha$, we have $\theta_i \geq \alpha$. Thus, we can then choose $\delta = n^{-1/2+\epsilon}$ and $\delta' = (\alpha \log n)^{-1/2+\epsilon}$. Since $p, q = \omega(\frac{\log^2 n}{n})$ from Assumption 4.3, by a simple union bound, we have for $\epsilon > 0$ small enough, for any $c > 0$ there is $C > 0$ such that

$$\mathbb{P}(U(n^{-1/2+\epsilon}, (\alpha \log n)^{-1/2+\epsilon})) \geq 1 - \frac{C}{n^c}. \tag{3}$$

Finally, we establish the lower bound for $\alpha$ indicated in the statement, which is $\frac{1}{\log n}$. The reason why we need this lower bound is that if $\alpha$ is too small, then the subgroups: $C_0(\alpha), C_1(\alpha)$ will be too sparse that their member nodes' degree is too small to assure the concentration inequalities.

By the definition of the event: $U(\delta, \delta')$ and union bound, we have

$$\mathbb{P}\left(\mathbf{I}_1(\delta)\right) \leq C\exp(-cn\delta^2)$$
$$\leq C\exp(-cn^{2\epsilon}) \qquad \text{(plug in } \delta = n^{-1/2+\epsilon})$$
$$\leq C/n^c \qquad \text{(if we choose } \epsilon \geq \frac{\log\log n}{2\log n} > 0),$$

and

$$\mathbb{P}\left(\bigcap_{i \in \mathcal{V}_\alpha} \mathbf{I}_{2,i}(\delta') \bigcap_{i \in \mathcal{V}_\alpha} \mathbf{I}_{3,i}(\delta')\right) \leq n \cdot C\exp(-cn(p+q)\alpha\delta'^2)$$
$$\leq n \cdot C\exp(-c\log^2 n \cdot \alpha\delta'^2)$$
$$\text{(by Assumption 4.3)}$$
$$= C\exp(\log n - c\log^2 n \cdot \alpha(\alpha\log n)^{-1+2\epsilon})$$
$$\text{(plug in } \delta' = (\alpha\log n)^{-1/2+\epsilon})$$
$$= C\exp(\log n - c\log n \cdot (\alpha\log n)^{2\epsilon})$$
$$= C\exp(\log n \cdot (1 - c \cdot (\alpha\log n)^{2\epsilon}))$$
$$= Cn^{1-c\cdot(\alpha\log n)^{2\epsilon}}.$$

We want to ensure the last term stays in $O(1/n^\beta)$ for some $\beta > 0$. Notice that if $\alpha < \log^{-1} n$, we cannot find suitable $\epsilon > 0$ for some small $c$ to satisfy $1 - c \cdot (\alpha\log n)^{2\epsilon} < 0$. Hence, we can conclude that a natural lower bound for $\alpha$ should be $\frac{1}{\log n}$, i.e., $\alpha > \frac{1}{\log n}$.

Thus, combining Equation 3 with this fact, we complete the proof.

$\qquad\qquad\qquad\qquad\qquad\qquad\qquad\qquad\qquad\qquad\qquad\qquad\qquad\qquad\qquad\qquad\qquad\quad$ $\square$

Next, in Lemma C.6, we claim that given the adjacency matrix $\mathbf{A}$, class memberships $(\epsilon_i)_{i \in [n]}$, degree-corrected factors $(\theta_i)_{i \in [n]}$ and a pre-defined threshold $\alpha > 0$, then with high probability, the convoluted node features $\tilde{\mathbf{x}}_i \approx \frac{p\boldsymbol{\mu}+q\boldsymbol{\nu}}{p+q}$ for $i \in C_0(\alpha)$ and $\tilde{\mathbf{x}}_i \approx \frac{q\boldsymbol{\mu}+p\boldsymbol{\nu}}{p+q}$ for $i \in C_1(\alpha)$.

**Lemma C.6** (Convoluted Feature Concentration). *Given* $\alpha \in (\frac{1}{\log n}, n]$, $C_0(\alpha), C_1(\alpha)$ *defined by Definition 4.1, and* $\mathcal{V}_\alpha = C_0(\alpha) \cup C_1(\alpha)$. *Conditionally on* $\mathbf{A}$, $(\epsilon_i)_{i \in [n]}$ *and* $(\theta_i)_{i \in [n]}$, *we have that for any* $c > 0$ *and some* $C > 0$, *with probability at least* $1 - \frac{C}{n^c}$, *for every node* $i \in \mathcal{V}_\alpha$ *and any unit vector* $\mathbf{w}$,

$$\left|\left\langle \tilde{\mathbf{x}}_i - \frac{p\boldsymbol{\mu}+q\boldsymbol{\nu}}{p+q}, \mathbf{w}\right\rangle (1+o(1))\right| = O\left(\sqrt{\frac{\log n}{dn(p+q)\alpha}}\right) \text{ for } i \in C_0(\alpha),$$

$$\left|\left\langle \tilde{\mathbf{x}}_i - \frac{q\boldsymbol{\mu}+p\boldsymbol{\nu}}{p+q}, \mathbf{w}\right\rangle (1+o(1))\right| = O\left(\sqrt{\frac{\log n}{dn(p+q)\alpha}}\right) \text{ for } i \in C_1(\alpha).$$

*Proof.* Since $(\mathbf{X}, \mathbf{A})$ is sampled from DC-CSBM$(\boldsymbol{\mu}, \boldsymbol{\nu}, p, q, \theta)$, when conditioning on $(\epsilon_i)_{i \in [n]}$, we have node $i$'s node feature $\mathbf{x}_i \sim \mathcal{N}(\boldsymbol{m}_i, \frac{1}{d}I)$ where $\boldsymbol{m}_i = \boldsymbol{\mu}$ if $i \in C_0$ and $\boldsymbol{m}_i = \boldsymbol{\nu}$ if $i \in C_1$. We can also write

$$\mathbf{x}_i = (1-\epsilon_i)\boldsymbol{\mu} + \epsilon_i\boldsymbol{\nu} + \frac{g_i}{\sqrt{d}},$$

where $g_i \sim \mathcal{N}(\mathbf{0}, \mathbf{I})$ is standard normal vector.

On the other hand, conditioning on the adjacency matrix $\mathbf{A}$ and class memberships $\epsilon = (\epsilon_i)_{i \in [n]}$, the mean of the convoluted feature of node $i$ can be written as

$$m(i) = \mathbb{E}[\tilde{\mathbf{x}}_i | \mathbf{A}, \epsilon] = \frac{1}{D_{ii}} \sum_{j \in [n]} \tilde{\mathbf{A}}_{ij} \boldsymbol{m}_j,$$

by the definition of the graph convolution operation ($\tilde{\mathbf{x}}_i = [\mathbf{D}^{-1}\tilde{\mathbf{A}}\mathbf{X}]_i$).

Thus, for any unit vector $\mathbf{w}$, we have

$$\tilde{\mathbf{x}}_i \cdot \mathbf{w} = \frac{1}{D_{ii}} \sum_{j \in [n]} \tilde{\mathbf{A}}_{ij}\langle \mathbf{x}_j, \mathbf{w}\rangle = \langle m(i), \mathbf{w}\rangle + \frac{1}{D_{ii}\sqrt{d}} \sum_{j \in [n]} \tilde{\mathbf{A}}_{ij} \cdot \langle g_j, \mathbf{w}\rangle. \tag{4}$$

Let us define $F_i = \frac{1}{D_{ii}\sqrt{d}} \sum_{j \in [n]} \tilde{\mathbf{A}}_{ij} \cdot \langle g_j, \mathbf{w}\rangle$ and observe that $\langle g_j, \mathbf{w}\rangle$ is a standard Gaussian random variable for all $j \in [n]$. Thus, we have that $F_i \sim \mathcal{N}(0, \frac{1}{dD_{ii}})$, conditioning on the adjacency matrix $\mathbf{A}$. Now we introduce Borell's inequality [2] to give a high-probability bound of $|F_i|$ for all $i \in \mathcal{V}_\alpha$.

**Lemma C.7** (Borell's Inequality, Theorem 2.1.1 in Adler et al. [2]). *Let $F_i \sim \mathcal{N}(0, \sigma_{F_i}^2)$ for each $i \in \mathcal{V}_\alpha$. Then for any $K > 0$, we have*

$$\mathbb{P}(\max_{i \in \mathcal{V}_\alpha} F_i - \mathbb{E}[\max_{i \in \mathcal{V}_\alpha} F_i] > K) \leq \exp(-\frac{K^2}{2\max_{i \in \mathcal{V}_\alpha} \sigma_{F_i}^2}).$$

We can further define the event $Q_\alpha = Q_\alpha(t) = \{\max_{i \in \mathcal{V}_\alpha} |F_i| \leq t\}$. Observe that

$$\begin{aligned}
\mathbb{P}(Q_\alpha^c | \mathbf{A}) &= \mathbb{P}(\max_{i \in \mathcal{V}_\alpha} |F_i| > t | \mathbf{A}) \\
&\leq 2\mathbb{P}(\max_{i \in \mathcal{V}_\alpha} F_i > t | \mathbf{A}) \\
&= 2\mathbb{P}(\max_{i \in \mathcal{V}_\alpha} F_i - \mathbb{E}[\max_{i \in \mathcal{V}_\alpha} F_i] > t - \mathbb{E}[\max_{i \in \mathcal{V}_\alpha} F_i] | \mathbf{A}).
\end{aligned}$$

If we let the union event $U := U(n^{-1/2+\epsilon}, (\alpha \log n)^{-1/2+\epsilon})$ defined the same as in Lemma C.4, then by Lemma C.7 and the definition of $\mathcal{V}_\alpha$,

$$\begin{aligned}
\mathbb{P}(Q_\alpha^c) &\leq \mathbb{P}(U \cap Q_\alpha^c) + \mathbb{P}(U^c) \\
&\leq 2\exp(-c'(t - \mathbb{E}[\max_{i \in \mathcal{V}_\alpha} F_i])^2 \cdot dn(p+q)\alpha) + \frac{1}{n^c},
\end{aligned}$$

for any $c > 0$ and some $c' > 0$.

By the definition of event $U(\delta, \delta')$, we can derive the upper bound of $\max_{i \in \mathcal{V}_\alpha} \sigma_{F_i}$ as follows:

$$\max_{i \in \mathcal{V}_\alpha} \sigma_{F_i} = \max_{i \in \mathcal{V}_\alpha} \sqrt{\frac{1}{dD_{ii}}} \leq \max_{i \in \mathcal{V}_\alpha} \sqrt{\frac{1}{d\frac{n}{2}(p+q)(1-\delta')\theta_i}} \leq \sqrt{\frac{2}{dn(p+q)(1-\delta')\alpha}}.$$

Since $\delta'$ is determined, we have for some constant $k', k > 0$,

$$\mathbb{E}[\max_{i \in \mathcal{V}_\alpha} F_i] \leq k'\sqrt{\log n} \max_{i \in \mathcal{V}_\alpha} \sigma_{F_i} \leq k\sqrt{\frac{\log n}{dn(p+q)\alpha}}.$$

By choosing $t = C'\sqrt{\frac{\log n}{dn(p+q)\alpha}}$ for some large constant $C' > k > 0$, we can obtain

$$t - \mathbb{E}[\max_{i \in \mathcal{V}_\alpha} F_i] \geq C'\sqrt{\frac{\log n}{dn(p+q)\alpha}} - k\sqrt{\frac{\log n}{dn(p+q)\alpha}} > 0.$$

Thus, we have

$$\begin{aligned}
\mathbb{P}(U \cap Q_\alpha) &\geq 1 - \mathbb{P}(U^c) - \mathbb{P}(Q_\alpha^c) \\
&\geq 1 - \frac{2}{n^c} - 2\exp(-c'(t - \mathbb{E}[\max_{i \in \mathcal{V}_\alpha} F_i])^2 dn(p+q)\alpha) \\
&\geq 1 - \frac{2}{n^c} - \frac{2}{n^{c'(C'-k)^2}}.
\end{aligned}$$

Recall that when conditioning on the event $U$, we have

$$m(i) = \frac{p\boldsymbol{\mu} + q\boldsymbol{\nu}}{p+q}(1 + o(1)) \quad \text{for } i \in C_0(\alpha), \tag{5}$$

$$m(i) = \frac{q\boldsymbol{\mu} + p\boldsymbol{\nu}}{p+q}(1 + o(1)) \quad \text{for } i \in C_1(\alpha). \tag{6}$$

Thus, on the event $U \cap Q_\alpha$, we have for each node $i \in \mathcal{V}_\alpha$,

$$|\langle \tilde{\mathbf{x}}_i - m(i), \mathbf{w}\rangle| = O\left(\sqrt{\frac{\log n}{dn(p+q)\alpha}}\right),$$

which completes the proof.

$\square$

Now we are ready to prove Theorem 4.4.

*Proof.* Recall the definition of linear separability, we need to find some unit vector $\mathbf{v} \in \mathbb{R}^d$ and $b \in \mathbb{R}$ such that

$$\langle \tilde{\mathbf{x}}_i, \mathbf{v}\rangle + b < 0 \quad \text{for } i \in C_0(\alpha),$$

$$\langle \tilde{\mathbf{x}}_i, \mathbf{v}\rangle + b > 0 \quad \text{for } i \in C_1(\alpha).$$

We fix $\tilde{\mathbf{v}} = \frac{1}{2\gamma}(\boldsymbol{\nu} - \boldsymbol{\mu})$ and $\tilde{b} = -\frac{\langle\boldsymbol{\mu}+\boldsymbol{\nu},\tilde{\mathbf{v}}\rangle}{2}$, where $\gamma = \frac{1}{2}\|\boldsymbol{\mu} - \boldsymbol{\nu}\|_2$. By Assumption 4.3, Lemma C.4 and C.6, with probability at least $1 - O(n^{-c})$ for any $c > 0$, for all $i \in C_0(\alpha)$, we have

$\langle \tilde{\mathbf{x}}_i, \tilde{\mathbf{v}}\rangle + \tilde{b}$

$\displaystyle = \frac{\langle p\boldsymbol{\mu} + q\boldsymbol{\nu}, \tilde{\mathbf{v}}\rangle}{p+q}(1 + o(1)) + O\left(\sqrt{\frac{\log n}{dn(p+q)\alpha}}\right) + \tilde{b}$ $\qquad$ (By Lemma C.6)

$\displaystyle = \left\langle \frac{(p-q)(\boldsymbol{\mu} - \boldsymbol{\nu})}{2(p+q)}, \tilde{\mathbf{v}}\right\rangle(1 + o(1)) + O\left(\sqrt{\frac{\log n}{dn(p+q)\alpha}}\right)$

$\displaystyle = -\gamma\Gamma(p,q)(1 + o(1)) + O\left(\sqrt{\frac{\log n}{dn(p+q)\alpha}}\right)$

$\displaystyle = -\gamma\Gamma(p,q)(1 + o(1)) + o(\gamma)$ $\qquad\qquad$ $\left(\alpha \in \omega\left(\frac{\log n}{dn(p+q)\gamma^2}\right)\right)$

$< 0.$ $\qquad\qquad$ (by Assumption 4.3)

Similarly, for all $i \in C_1(\alpha)$, we have

$\langle \tilde{\mathbf{x}}_i, \tilde{\mathbf{v}}\rangle + \tilde{b}$

$\displaystyle = \frac{\langle q\boldsymbol{\mu} + p\boldsymbol{\nu}, \tilde{\mathbf{v}}\rangle}{p+q}(1 + o(1)) + O\left(\sqrt{\frac{\log n}{dn(p+q)\alpha}}\right) + \tilde{b}$ $\qquad$ (By Lemma C.6)

$\displaystyle = \left\langle \frac{(p-q)(\boldsymbol{\nu} - \boldsymbol{\mu})}{2(p+q)}, \tilde{\mathbf{v}}\right\rangle(1 + o(1)) + O\left(\sqrt{\frac{\log n}{dn(p+q)\alpha}}\right)$

$\displaystyle = \gamma\Gamma(p,q)(1 + o(1)) + O\left(\sqrt{\frac{\log n}{dn(p+q)\alpha}}\right)$

$\displaystyle = \gamma\Gamma(p,q)(1 + o(1)) + o(\gamma)$ $\qquad\qquad$ $\left(\alpha \in \omega\left(\frac{\log n}{dn(p+q)\gamma^2}\right)\right)$

$> 0.$ $\qquad\qquad$ (by Assumption 4.3)

The above two inequalities imply the linear separability of $\{\tilde{\mathbf{x}}_i, i \in \mathcal{V}_\alpha\}$, which completes the proof. $\square$

Table 3: Description of parameters used in the controlled experiments.

| Parameter Name | Description |
|---|---|
| $n$ | Number of vertices in the graph. |
| cluster size slope | the slope of cluster sizes when ordered by size. |
| feature dimension | the number of dimensions of node features. |
| feature center distance | distance between feature cluster centers. |
| $p/q$ ratio | the ratio of intra-class edge probability to inter-class edge probability. |
| average degree | the average expected degrees of nodes. |
| power exponent | the value of power-law exponent used to generate expected node degrees. |
| feature cluster variance | variance of feature clusters around their centers. |

Table 4: Remaining parameters used in experiments. One of the parameters is manipulated to generate synthetic datasets with varying data properties indicated in the first column.

| Experiments | $p/q$ ratio | Average Degree | Power Exponent | Feature Cluster Variance |
|---|---|---|---|---|
| *Gini-Degree* | 3 | 20 | [1.5, 2, 2.5, 3, 5] | 0.25 |
| *Average Degree* | 3 | [10, 20, 30, 40, 50] | 2 | 0.25 |
| *Edge Homogeneity* | [1,2,3,5,10] | 20 | 2 | 0.1 |
| *In-Feature Similarity* | 2 | 20 | 2 | [2, 1, 0.5, 0.2, 0.1] |
| *Feature Angular SNR* | 2 | 20 | 2 | [2, 1, 0.5, 0.2, 0.1] |

# D Controlled Experiments of Identified Salient Factors

From Section 3.3, we discover six prominent dataset properties that correlate with some or all of the GNN models' performance. In Section 5, we have presented controlled experiments for Gini-Degree to verify its relationship to GNNs' performance (Table 2). In this section, we further conduct controlled experiments for all the remaining identified salient factors, except for *Pseudo Diameter*, which is hard to control via manipulating explicit parameters provided by GraphWorld.

Across all experiments, we fix the number of nodes $n = 5000$, cluster size slope as $0.0$, the number of clusters as $4$, feature dimension as $16$, and feature center distance as $0.05$. For each of the experiments, we will keep most of the remaining GraphWorld parameters the same and only vary one of the parameters. The remaining parameters that we will manipulate are the $p/q$ ratio, average degree, power exponent, and feature cluster variance. We give a short description of all the parameters in Table 3. For completeness, we summarize the value of the remaining parameters used in all four experiments in Table 4.

Table 5, 6, and 7 show the results of the four controlled experiments, correspondingly. Note that varying feature cluster variance can manipulate *In-Feature Similarity* and *Feature Angular SNR* simultaneously (Table 7). In general, all the results closely follow the regression results indicated in Table 1 and the discussion in Section 3.3.

Table 5: Controlled experiment results for varying *Average Degree*. Standard deviations are derived from 5 independent runs. The performances of all models except for GAT, MixHop, and MLP have an evident positive correlation with *Average Degree*.

| *Average Degree* | GCN | GAT | GraphSAGE | MoNet | MixHop | LINKX | MLP |
|---|---|---|---|---|---|---|---|
| 10 | 0.71±0.018 | 0.67±0.009 | 0.725±0.002 | 0.556±0.024 | 0.696±0.001 | 0.693±0.006 | 0.632±0.004 |
| 20 | 0.823±0.001 | 0.734±0.013 | 0.797±0.006 | 0.593±0.012 | 0.806±0.003 | 0.825±0.001 | 0.54±0.024 |
| 30 | 0.839±0.005 | 0.722±0.017 | 0.801±0.002 | 0.761±0.005 | 0.756±0.002 | 0.852±0.003 | 0.653±0.004 |
| 40 | 0.876±0.003 | 0.742±0.006 | 0.825±0.001 | 0.795±0.002 | 0.794±0.003 | 0.876±0.002 | 0.648±0.003 |
| 50 | 0.9±0.004 | 0.734±0.019 | 0.86±0.002 | 0.814±0.003 | 0.788±0.011 | 0.89±0.005 | 0.651±0.002 |

Table 6: Controlled experiment results for varying *Edge Homogeneity*. Standard deviations are derived from 5 independent runs. The performances of all models except for MixHop and MLP have an evident positive correlation with *Edge Homogeneity*.

| Edge Homogeneity | GCN | GAT | GraphSAGE | MoNet | MixHop | LINKX | MLP |
|---|---|---|---|---|---|---|---|
| 0.249 | 0.737±0.004 | 0.565±0.009 | 0.732±0.005 | 0.515±0.004 | 0.836±0.002 | 0.823±0.005 | 0.744±0.033 |
| 0.375 | 0.873±0.002 | 0.825±0.011 | 0.847±0.003 | 0.57±0.009 | 0.945±0.002 | 0.93±0.003 | 0.93±0.001 |
| 0.452 | 0.917±0.002 | 0.887±0.004 | 0.896±0.007 | 0.598±0.005 | 0.947±0.001 | 0.949±0.002 | 0.784±0.09 |
| 0.559 | 0.925±0.002 | 0.89±0.004 | 0.925±0.004 | 0.678±0.003 | 0.913±0.005 | 0.943±0.005 | 0.9±0.004 |
| 0.702 | 0.946±0.004 | 0.933±0.004 | 0.953±0.001 | 0.802±0.003 | 0.942±0.001 | 0.959±0.001 | 0.865±0.004 |

Table 7: Controlled experiment results for varying *In-Feature Similarity / Feature Angular SNR*. Standard deviations are derived from 5 independent runs. The performances of all models except for MoNet have an evident positive correlation with *In-Feature Similarity / Feature Angular SNR*.

| In-Feature Similarity | Feature Angular SNR | GCN | GAT | GraphSAGE | MoNet | MixHop | LINKX | MLP |
|---|---|---|---|---|---|---|---|---|
| 0.506 | 1.009 | 0.478±0.016 | 0.412±0.016 | 0.446±0.005 | 0.562±0.021 | 0.433±0.001 | 0.598±0.002 | 0.402±0.002 |
| 0.516 | 1.022 | 0.563±0.004 | 0.47±0.006 | 0.517±0.008 | 0.615±0.002 | 0.531±0.003 | 0.661±0.004 | 0.47±0.001 |
| 0.527 | 1.039 | 0.717±0.008 | 0.507±0.006 | 0.6±0.006 | 0.555±0.021 | 0.621±0.007 | 0.737±0.001 | 0.486±0.003 |
| 0.582 | 1.101 | 0.784±0.011 | 0.599±0.014 | 0.74±0.01 | 0.533±0.01 | 0.848±0.001 | 0.854±0.001 | 0.611±0.003 |
| 0.602 | 1.154 | 0.887±0.006 | 0.791±0.004 | 0.825±0.006 | 0.627±0.004 | 0.924±0.004 | 0.913±0.006 | 0.915±0.002 |

# E  Robustness of the Sparse Regression Analysis

To demonstrate the robustness of the identified salient factors (defined in Section 3.3), we expand our sparse regression analysis to include five additional models that are more recent and popular. The models are TAGCN [10], GATv2 [5], SGC [40], APPNP [12] and GCNII [7]. For the additional experiments, we adopt the same configuration as in Appendix B. The updated analysis result is presented in Table 8.

We can observe that the widely influential factors and the narrowly influential factors all remain salient after incorporating the five additional models. The results show that our proposed multivariate regression analysis is robust with respect to our diverse choice of GNN models.

Table 8: The estimated coefficient matrix **B** of the multivariate sparse regression analysis with five additional GNN models. These five models are indicated in blue. Each entry indicates the strength (magnitude) and direction ($+, -$) of the relationship between a graph data property and the performance of a GNN model. The six most salient data properties are indicated in **bold**. Notice that these factors are the same as the ones we presented in Section 3.3.

| Graph Data Property | GCN | GAT | GraphSAGE | MoNet | MixHop | LINKX | MLP | TAGCN | GATv2 | SGC | APPNP | GCNII |
|---|---|---|---|---|---|---|---|---|---|---|---|---|
| Edge Density | 0 | 0 | 0 | 0 | 0 | 0.0279 | 0.0937 | 0 | 0 | 0 | 0 | 0 |
| **Average Degree** | 0.2136 | 0 | 0.098 | 0.1047 | 0 | 0.3362 | 0 | 0.173 | 0 | 0.4588 | 0 | 0 |
| **Pseudo Diameter** | 0 | -0.3824 | -0.1608 | -0.0173 | -0.4915 | -0.3937 | -0.6191 | -0.2514 | -0.1428 | -0.0816 | -0.401 | -0.2962 |
| Degree Assortativity | 0 | 0 | 0 | -0.0587 | 0 | 0 | 0 | 0 | 0 | 0 | 0 | 0 |
| RSLCC | 0.1014 | 0 | 0 | 0.0673 | 0 | 0.1312 | 0 | 0.0333 | 0 | 0 | 0 | 0 |
| ACC | 0 | 0 | 0 | 0 | 0 | 0 | -0.0523 | -0.1139 | 0.0276 | 0 | 0 | 0 |
| Transitivity | 0 | -0.0458 | 0 | -0.148 | 0 | 0.2168 | 0 | 0 | 0 | -0.0795 | 0 | -0.0315 |
| Degeneracy | 0 | 0 | 0 | 0 | 0 | 0 | -0.1555 | 0 | -0.0652 | -0.3099 | -0.0276 | 0 |
| **Gini-Degree** | -0.4437 | -0.2955 | -0.3313 | -0.292 | -0.4269 | -0.3681 | -0.1993 | -0.3838 | -0.2043 | -0.1907 | -0.3021 | -0.33 |
| **Edge Homogeneity** | 0.714 | 0.4197 | 0.7241 | 0.8108 | 0.6396 | 0.2017 | 0.4777 | 0.7147 | 0.7007 | 0.2817 | 0.7962 | 0.7184 |
| **In-Feature Similarity** | 0.3103 | 0.0926 | 0.1878 | 0.0989 | 0.4576 | 0.6406 | 0.2421 | 0.4394 | 0.0359 | 0 | 0 | 0.0255 |
| Out-Feature Similarity | 0 | 0 | 0 | 0 | 0 | 0 | 0 | 0 | 0 | 0 | 0 | 0 |
| **Feature Angular SNR** | 0.2492 | 0.0393 | 0.2455 | 0 | 0.2355 | 0.3564 | 0.3682 | 0.1354 | 0.3308 | -0.0997 | 0.2733 | 0.359 |
| Homophily Hat | 0 | 0.4569 | 0 | 0 | 0 | 0 | 0 | 0 | 0 | 0.4795 | 0 | 0 |
| Attribute Assortativity | 0 | 0 | 0 | 0 | 0 | 0 | 0 | 0 | 0 | 0 | 0 | 0 |

