# OpenReview forum: "A Metadata-Driven Approach to Understand Graph Neural Networks"
_NeurIPS.cc/2023/Conference — NeurIPS 2023 poster_

### Official Review · Reviewer_zbzi · 2023-07-04

**Soundness:** 4 excellent
**Presentation:** 4 excellent
**Contribution:** 3 good
**Rating:** 7
**Confidence:** 3

**Summary:**

Edit: Based on the new experiments, I am increasng my score slightly

They conduct a multivariate regression on the metadata of datasets and performance of multiple GNN models and examine the correlation between GNN performance and data properties.

They show that Gini-Degree, Edge Homogeneity, and In-feature similarity are salient features for all of the models they consider and that average degree, pseudo-diameter, and feature angular snr are salient for some of the models.

They also perform some analysis on how the gini-coefficient of the degree distribution affects model performances on a novel SBM.

Overall, I think this is a novel approach to an important problem and a good paper.

**Strengths:**


This is a novel approach to an important problem. Advances in ML are based on empirical performance on benchmarks as much, if not more than, as they are on theoretical guarantees. Therefore, understanding the benchmark-datasets, and what makes a network preform well on them is of paramount importance. This paper which focuses on the metadata of the datasets can play a complementary role to other model driven analyses.

The DC-SBM model is interesting as is the concept of separate the $C(\alpha)'s$ rather than the entire classes.

As a disclaimer: I did not read the proof of Theorem 4.4 as thoroughly as I would like to due to time constraints. At a glance, it appears to be a well-organized application of concentration of measure techniques. Would be curious if any of the reviewers have checked it more closely. (This is obviously not the authors fault and I do not hold it against them.)

**Weaknesses:**


The characterization of ``universal factors" is somewhat misleading as they are only universal amongst the several GNN models considered. Indeed, a potential direction, building off of this work would be to find GNN models which are not affected by these `universal' factors. For instance, there are GNNs designed to alleviate the oversmoothing issue and they might be more robust to datasets with low edge homogeneity or in-feature similarity.

Some of the equations in the proof are spilling off of the page.


**Questions:**


There are GNNs which use non-local aggregations such [1] and [2]. Do you think that they would be able to avoid the dependence on the Pseudo Diameters?

[1] https://arxiv.org/abs/2305.13084
[2] https://arxiv.org/abs/2201.08932

---

> ### Author Rebuttal · Authors · 2023-08-09
>
> We appreciate the reviewer's acknowledgment of the novelty and importance of this study. We address your concerns and questions in detail below.
>
> **W1: [Characterization of universal factors.]**
>
> A: This is a good point. These so-called "universal" factors are only salient for the GNN models we considered in the analysis. We agree that "universal" might be misleading. We plan to change the names to "widely influential factors" and "narrowly influential factors".
>
> **W2: [Equations spilling off the page.]**
>
> A: Thank you for the comment. We will fix them in our next version.
>
> **Q1: [GNNs with non-local aggregations vs. Pseudo Diameter.]**
>
> A: Thank you for the suggestion. We conducted additional experiments to investigate this question. Due to time and GPU memory constraints, we only performed experiments on the first paper’s model (FLODE) mentioned by the reviewer.
>
> We tried two versions of the regression analysis to investigate whether there is a dependency between FLODE’s performance and Pseudo Diameter. In the first version, we only use FLODE’s performance as regression targets (univariate regression analysis). In the second version, we use the performance of all models as regression targets (multivariate regression analysis).
>
> We agree with your intuition that since fLode uses non-local aggregation to alleviate over-smoothing problems, its performance should not be very sensitive to the diameter of the graph. However, the results from both regression analyses show that there is still a dependency between fLode performance and Pseudo Diameter. We do not have a good explanation for this result yet. This will undoubtedly be a very intriguing follow-up work to our current paper.
>
> The regression results (coefficients of FLODE) from univariate/multivariate regression analysis are shown below.
> |  Graph Data Property | Coefficient from Univariate Analysis  | Coefficient from Multivariate Analysis |
> |:----: | :----: | :----: |
> | Edge Density |  0.1676 | 0.1599 |
> | Average Degree  | 0  | 0|
> | Pseudo Diameter  | -0.2263  | -0.2348|
> | Degree Assortativity |  0 | 0|
> | RSLCC| 0  | 0.0566|
> | ACC |  0 | 0|
> | Travsitivity| 0.3203  | 0.3614|
> | Degeneracy |  0| 0 |
> | Gini-Degree|  -0.2271 | -0.2597|
> | Edge Homogeneity | 0.8022 | 0.8128|
> | In-Feature Similarity | 0  | 0|
> | Out-Feature Similarity|  0 | 0|
> | Feature Angular SNR| 0.3158  | 0.3277|
> | Homophily Measure | 0  | 0.031|
> | Attribute Assortativity | 0  | 0|

---

> > ### Author Response · Authors · 2023-08-17
> > **Thank you for acknowledging our response!**
> >
> > Thank you for acknowledging our response and raising the evaluation score!

---

### Official Review · Reviewer_FJsn · 2023-07-04

**Soundness:** 4 excellent
**Presentation:** 3 good
**Contribution:** 2 fair
**Rating:** 5
**Confidence:** 5

**Summary:**

This paper offers an intriguing analysis aimed at analyzing the performance of Graph Neural Networks (GNNs). The analysis explores various graph-theoretic properties of the graph datasets, which may impact the performance of GNNs. The authors investigate 15 graph properties across 15 datasets and examine the coefficient matrix of a multivariate linear regression model using the obtained metadata. The analysis reveals an intriguing observation: a negative correlation between Gini-Degree and the performance of GNNs. The authors support this observation with theoretical analysis and controlled experiments, providing compelling evidence to support their claim.

**Strengths:**

1 - the hypothesis considered makes sense and properly evaluated
2 - the paper is well-written and well-organized
3 - the investigation reveal clear results and are supported by theoretical analysis and controlled experiments

**Weaknesses:**

1 - I appreciate the thorough investigation of the proposed hypothesis on generated synthetic datasets for node classification. However, it would be intriguing to expand the analysis by exploring a few real-world datasets that demonstrate both high and low Gini-degree. This would provide valuable insights into the applicability and generalizability of the observed relationship in real-world scenarios.



**Questions:**

1- I am curious to know if this observation holds true for the graph classification task as well.

2- Table 2 indicates that all GNNs exhibit improved performance with a decrease in Gini-degree. However, is this consistently the case, or is there a relationship with the type of convolution used? It would be intriguing to explore the connection between different graph convolution operators such as MPNN and GIN, and their relationship with Gini-Degree.

**Limitations:**

-

---

> ### Author Rebuttal · Authors · 2023-08-09
>
> We appreciate the reviewer's acknowledgment of the soundness and clarity of our paper. We address your individual concerns in detail below.
>
> **W1: [Real-world datasets with varying Gini-Degree.]**
>
> A: We would like to clarify that the real-world datasets used in Section 3 do have varying Gini-Degree. Specifically, these real-world datasets have their Gini-Degree ranging from 0.4 to 0.73. If there is no variation of Gini-Degree among the datasets, the sparse penalty will likely drive the coefficient for this factor to 0.
>
> However, it is tricky to conduct controlled experiments on real-world datasets, as it is difficult to find datasets that only vary in Gini-Degree while all other factors are similar.
>
> **Q1: [Graph classification tasks.]**
>
> A: We consider extending our current framework to analyze other graph learning tasks, including graph classification tasks. But notice that for graph classification tasks, the datasets become “sets of graphs” and our regression analysis only works for a single graph for each dataset.
>
> Intuitively, we can compute the average graph properties among graphs in a single dataset to find a representative set of factors describing the entire graph classification dataset. And it will be straightforward to apply the same regression analysis to discover prominent factors.
>
> Unfortunately, we didn’t have time to run multiple graph classification algorithms and conduct a thorough analysis for this task in the rebuttal phase. It will be an exciting future direction for further research, including defining other aggregation schemes other than simple averaging.
>
> **Q2: [Consistent for different types of convolution.]**
>
> A: In our paper, we observe that it is consistent for a few different types of convolutional operations, such as GCN, GAT, MixHop, etc. To further verify the consistency and robustness of our analysis, during the rebuttal period, we incorporate five more models that are popular on node classification tasks, including TAGCN, GATv2, SGC, APPNP, and GCNII, into our multivariate regression analysis. The regression results are shown in the global pdf. We can observe that the factors we discover in Section 3.3 remain valid even if we involve these additional GNN models.

---

> > ### Comment · Reviewer_FJsn · 2023-08-16
> > **Response to Rebuttal**
> >
> > The authors have effectively addressed my comments. I do not have any further comments, and my score will remain unchanged.

---

> > > ### Author Response · Authors · 2023-08-17
> > > **Thank you for acknowledging our response!**
> > >
> > > Thank you for reading our response and acknowledging that we have effectively addressed your concerns!

---

### Official Review · Reviewer_f4if · 2023-07-06

**Soundness:** 2 fair
**Presentation:** 3 good
**Contribution:** 2 fair
**Rating:** 5
**Confidence:** 3

**Summary:**

Graph neural networks (GNNs) have demonstrated significant success across various problem areas. In this study, the authors propose a metadata-driven approach to examine the sensitivity of GNNs to graph data properties, spurred by the growing availability of graph learning benchmarks. A multivariate sparse regression analysis is performed on GNN performance across a range of datasets, identifying a set of significant data properties. The authors then focus on the degree distribution, investigating its influence on GNN performance through theoretical analysis and controlled experiments.

**Strengths:**

1. The paper is well-written and easy to comprehend.
2. The concept of metadata-driven analysis is intriguing and holds potential for further refinement in future research.
3. The paper includes a theoretical contribution that enhances our understanding of GNNs' success.

**Weaknesses:**

1. Given the comprehensive nature of this study, the authors should justify their selection of the GNN models used in Section 3, given the hundreds of models available in this field. There should be a principled approach to choosing models that are (1) representative, (2) popular, and (3) have diverse operational mechanisms, ensuring that the study is reliable and generalizable.
2. The metadata-driven analysis presented in Section 3 is somewhat incomplete. As mentioned by the authors in Lines 201 - 202, Section 3 only investigates the correlation between GNN performances and data properties. Causality is explored later and exclusively for Gini-Degree. The insights related to edge homogeneity and in-feature similarity are primarily replications of existing work findings. Regarding the other three properties, the authors note that "we do not yet have a good understanding of the mechanism of how these data properties impact model performance," indicating that the study is not fully complete.
3. There should be a more comprehensive discussion relating to node degrees, as this is a popular property used to understand graphs and networks. The authors claim that "the finding of the Gini coefficient of the degree distribution" is novel, but how is it distinctly novel compared to existing work? I suggest the authors connect the findings in this work more closely to those in previous studies to strengthen their theoretical contribution.

**Questions:**

The overall contribution of the study seems weak, mainly because of Section 3. Please refer to the weaknesses listed above.

**Limitations:**

The authors have acknowledged the limitations of their approach.

---

> ### Author Rebuttal · Authors · 2023-08-09
>
> We appreciate the reviewer's acknowledgment of the clarity of our paper, the value of our proposed metadata-driven analysis, and our theoretical contribution. We address your concerns in detail below.
>
> **W1: [Selection of the GNN model we use for regression analysis.]**
>
> A: We agree that many possible GNN models can be used for the regression analysis. Our choice inherited the ones selected by the Graph Learning Indexer (GLI) paper [1] since we rely on this library to obtain the metadata.
>
> [1] Ma et al. Graph Learning Indexer: A Contributor-Friendly and Metadata-Rich Platform for Graph Learning Benchmarks. 2022
>
> To address your concern, we further implement five additional models (that are popular and widely cited) to obtain more metadata for the regression analysis, including SGC, APPNP, TAGCN, GATv2, and GCNII. The result of the regression analysis can be found in the global pdf we submitted with our rebuttal.
>
> The six salient factors we discover in our paper remain crucial after including the metadata associated with these five models, demonstrating the robustness and efficacy of our regression analysis and the salient factors we found.
>
> **W2: [Incompleteness of our regression analysis.]**
>
> A: The metadata-driven analysis shows the saliency of the correlation between certain dataset properties and GNN model performances. Since correlation does not necessarily imply causality, we further explore the relationship between Gini-Degree and GNN performance to validate the observation made in Section 3.
>
> We chose to investigate Gini-Degree because edge homogeneity and in-feature similarity are well-known factors. There are various papers that verify these factors are essential for GNNs through theoretical and empirical analysis. Hence, edge homogeneity and in-feature similarity are not significant findings of our analysis. This discovery is more like verifying that our regression analysis is valid and can uncover widely known and studied factors. On the other hand, degree distribution is widely considered an essential factor but lacks in-depth analysis. We provide a thorough theoretical analysis to explain the relationship between Gini-Degree and GNN performances.
>
> It is true that we do not have a technical justification for the three selective factors (Average Degree, Pseudo Diameter, and Feature Angular SNR) mentioned in our paper. However, this does not undermine the effectiveness and completeness of our metadata-driven analysis, which serves as our prime contribution. Further investigation for these selective factors will certainly be possible direction building off this work, but this will not affect the fact that this paper can complement current data-driven analysis approaches.
>
> **W3: [Discussion relating to node degrees.]**
>
> A: We have included some GNN works discussing node degrees in Section 2.3. Previous works on this topic recognize that node-wise classification accuracy will vary depending on node degrees. But most of them designed new GNN architecture to overcome the low accuracy on lower degree nodes. Our theoretical contribution laid a more fundamental basis for a dataset-level understanding. We utilized a simplified version of GCN to tell the reason why degree distribution has a direct relationship with GNN performance.

---

> > ### Comment · Reviewer_f4if · 2023-08-14
> > **Thank you for the response**
> >
> > I appreciate the authors response. This paper effectively serves as a proof-of-concept in elucidating the impact of data properties on a model's performance. While there are some practical limitations, I have increased my score.

---

> > > ### Author Response · Authors · 2023-08-17
> > > **Thank you for acknowledging our response!**
> > >
> > > Thank you for acknowledging our response and raising the evaluation score!

---

### Official Review · Reviewer_aoxj · 2023-07-06

**Soundness:** 2 fair
**Presentation:** 3 good
**Contribution:** 2 fair
**Rating:** 4
**Confidence:** 4

**Summary:**

The authors analyze the sensitivity of GNNs to graph properties such as degree distribution, homophily, etc. Using a multivariate sparse regression analysis (based on Lasso), they find a set of properties that are universal in that they effect all kinds of GNNs and some properties that are local in that they effect only a few of the GNNs. Among the global properties, one that is prominent is degree distribution that has an inverse effect on node classification accuracy. In the second half of the paper, the authors analyze the reasons behind this behavior using a theoretical model based on a Degree-Corrected Contextual Stochastic Block Model. The results of the model are validated by a controlled experiment.


**Strengths:**

The paper presents a good analysis of the effect of degree distribution through empirical and theoretical analysis.


**Weaknesses:**

Translating the theoretical results into empirical outcomes requires the total degree to be bounded so that there are many nodes with similar degrees. Is this true for the datasets the authors considered?

At the same time, it is not surprising that it becomes harder to classify nodes that have similar degrees since one of the important features is not useful anymore.

Can the theoretical model explain the other universal features such as edge heterogeneity and in-feature similarity?

Lasso requires the errors to be Gaussian. Did the authors verify this?

It is interesting to see that MLP also suffers from the effect of degree distribution.

Overall, the impact of the paper is not likely to be high since it only considers the problem of node classification.

**Questions:**

Please see above.

**Limitations:**

None.

---

> ### Author Rebuttal · Authors · 2023-08-09
>
> We appreciate the reviewer's acknowledgment that we present a good analysis of the effect of the degree distribution. We address your concerns in detail below.
>
> **W1: [Requirement of bounded total degree for datasets.]**
>
> A: We clarify that a bounded total degree does not necessarily imply that many nodes will have similar degrees. Given the total degree fixed, how “equally” the edges are distributed among nodes is controlled by Gini-Degree. There could still be a skewed degree distribution for some choice of Gini-Degree. We believe that the bounded degree assumption is not a strong assumption.
>
> **W2: [Node degree as an important feature for node classification.]**
>
> A: Indeed, a higher Gini-Degree implies that the graph will have more nodes with similar node degrees. However, our main contribution is to verify that a higher Gini-Degree will result in poorer performance of GCNs. Under this situation, fewer nodes will have higher degrees in this dataset. By the assumption we made in Section 4, we know that nodes with lower degrees will be harder to classify, and hence the GCN model will perform worse in this case. (According to the discussion in Section 4.4)
>
> So the main reason why datasets with high Gini-Degree are considered more challenging tasks is not brought by the fact that we lose “node degree” as a node feature.
>
> Moreover, Node degrees should not be considered a crucial feature in general. For example, if the dataset is generated from a DC-CSBM (in Section 4.2), nodes with larger degrees will be easier to classify in our setting, even though they have very similar degrees.
>
>
> **W3: [Can the theoretical model explain other universal features?]**
>
> A: We clarify that the purpose of our theoretical analysis is NOT to explain all the factors discovered in our regression analysis in the first place. Instead, both the theoretical analysis (Section 4) and the controlled experiment (Section 5) are designed to explain the impact of Gini-Degree, as a proof-of-concept verification that our main contribution, the metadata-driven approach for discovering salient graph characteristics (Section 3), is effective.
>
> It is indeed also interesting to further understand other salient factors discovered by our metadata-driven approach, but we believe that examining all factors is beyond the scope of this paper and will leave it for future research.
>
>
> **W4: [Lasso requires the errors to be Gaussian.]**
>
> A: We would like to point out that Lasso has been widely used for the purpose of feature selection, which works without requiring the data to have Gaussian errors. In fact, the effectiveness of Lasso for feature selection has been theoretically verified under weak conditions when Gaussian assumptions are not met. See, for example, the following paper [1].
>
> [1] Zhao and Yu, On Model Selection Consistency of Lasso, 2006.
>
> **W5: [MLP suffers from degree distribution.]**
>
> In fact, MLP is much less influenced by degree distribution. In Table 1, the absolute magnitude of the coefficient for MLP is much smaller than that for other models. In Table 2, there is not much variation of MLP performance for varying Gini-Degree.
>
> **W6: [Limited impact due to node classification problems only.]**
>
> A: While we acknowledge that our empirical study focuses on node classification datasets in this paper, we would like to point out that the conceptual framework of a metadata-driven approach is not limited to node classification tasks. Furthermore, node classification is already a broad class of problems. So we respectfully disagree with the reviewer's comment that a paper would have limited impact if "it only considers the problem of node classification". In addition, we believe that a side product of our proposed approach, the understanding of the impact of degree distribution on GNN performance, is also a significant contribution in its own right.

---

> ### Comment · Reviewer_aoxj · 2023-08-16
> **Thanks for the response**
>
> I than the authors for their response. I have increased my score to 4. I still think the paper has limited novelty for reasons outlined earlier.

---

> > ### Author Response · Authors · 2023-08-17
> > **Thank you for acknowledging our response!**
> >
> > We thank the reviewer for acknowledging our response and raising the evaluation score. While we believe that our responses have addressed the concerns you've previously raised, we genuinely value your feedback and aim to improve our manuscript based on it.
> >
> > To ensure we can make the necessary changes and clarifications, **could you kindly specify which reasons outlined earlier have not been effectively addressed by our response?** This will allow us to more directly address these concerns and further enhance the quality and novelty of our work. Thank you!

---

> > > ### Comment · Reviewer_aoxj · 2023-08-18
> > >
> > > Weaknesses W2 and W6.
> > > I think the paper will have limited impact because node classification has been studied a lot and the impact of degree distribution is also understood. The novelty is not much.

---

> > > > ### Author Response · Authors · 2023-08-18
> > > > **Clarification on some potential misinterpretations**
> > > >
> > > > We appreciate the reviewer's reply.
> > > >
> > > > We would like to emphasize that the primary objective of our paper is not merely about node classification or the influence of degree distribution. Instead, our main contribution lies in the introduction of a novel metadata-driven analysis designed to discern how varying data characteristics can impact the performance of GNN models. Crucially, this general methodology extends beyond the realm of node classification.
> > > >
> > > > To verify the effectiveness of the proposed metadata approach, we have chosen node classification datasets as our testbed. We would like to highlight that 1) **the fact that node classification tasks are so popular and broadly relevant to real-world applications makes them an ideal testbed**; 2) the proposed metadata approach is a novel perspective to understand existing GNN models, rather than a new graph learning method for node classification tasks.
> > > >
> > > > Additionally, the theoretical and empirical analyses of Gini-Degree serve as a proof-of-concept verification that the proposed metadata-driven analysis can effectively identify salient factors. The salience of the other two major factors, edge homogeneity and in-feature similarity, have been well-established in the literature, which is why we focus on Gini-Degree in the second half of the paper. **Even if the salience of degree distribution had been well-understood, it still justifies the effectiveness of our metadata approach**. However, to our best knowledge, there has not been any previous study that investigates how the Gini coefficient of the degree distribution impacts GNN performance. So both our analysis and results are novel.
> > > >
> > > > Lastly, feedback from other reviewers further validates that **our main contribution is the novel metadata-driven analysis as opposed to a study of node classification problems**. For instance, Reviewer f4if mentioned, "The concept of metadata-driven analysis is intriguing", and applauded our work as an effective "proof-of-concept in elucidating the impact of data properties on a model's performance." Similarly, Reviewer zbzi lauded our innovative approach and noted how our focus on dataset metadata can complement other model-driven analyses.
> > > >
> > > > **In summary, the main focus of this paper is a novel metadata-driven analysis to understand GNNs, as opposed to a study of node classification problems; the analysis of the degree distribution mainly serves as a proof-of-concept verification for the effectiveness of the proposed metadata-driven methodology; and the specific theoretical results on the Gini coefficient of degree distribution are also novel.**

---

### Author Rebuttal · Authors · 2023-08-09

Dear Reviewers,

We appreciate your thorough evaluation of our paper and the valuable feedback you have provided. We have carefully considered your comments and responded individually. We would also like to summarize below our response to the major concerns raised in the reviews.

*Clarification of Main Contribution.* The primary focus of our work is to introduce a metadata-driven analysis approach that sheds light on the relationship between complex dataset properties and the performance of GNNs. We want to clarify that the theoretical analysis and controlled experiments were not designed to verify every single salient factor comprehensively, but rather to serve as proof-of-concept verification for the proposed metadata-driven methodology. We chose the factor related to degree distribution for further investigation mainly due to its popularity and the lack of in-depth studies in previous works. Theoretical framework or experimental setup for comprehensively verifying all salient factors identified by our methodology are exciting future directions.

*Robustness of analysis.* As suggested by reviewers, we have expanded our analysis to include five more models that are more recent and also popular. The experiment results are reported in the global pdf attached with this general response. The results show that our proposed multivariate regression analysis is fairly robust with respect to the choice of GNN models. For this additional experiment, we adopt the same configuration as our paper for the default training settings.

We would like to thank all the reviewers again for their valuable feedback, which has enabled us to clarify our central contributions and showcase the robustness of the metadata-driven regression analysis. We hope our responses have successfully addressed all the concerns and look forward to hearing from the reviewers for further comments.

---

### Decision · Program_Chairs · 2023-09-21

**Decision:**

Accept (poster)

**Comment:**

This paper proposes a metadata-driven approach to analyze the sensitivity of GNNs to graph data properties, motivated by the increasing availability of graph learning benchmarks. The authors perform a multivariate sparse regression analysis on the metadata derived from benchmarking GNN performance across diverse datasets, yielding a set of salient data properties.

Overall, the reviewers have acknowledged that concept of metadata-driven analysis is intriguing and holds potential for further refinement in future research, although there are concerns about justifying selection of the GNN models (addressed in rebuttal), more thorough comparison with existing works, and some practical limitations.